# Investigating the DNA-Binding Site for VirB, a Key Transcriptional Regulator of *Shigella* Virulence Genes, Using an In Vivo Binding Tool

**DOI:** 10.3390/genes10020149

**Published:** 2019-02-15

**Authors:** Monika M.A. Karney, Joy A. McKenna, Natasha Weatherspoon-Griffin, Alexander D. Karabachev, Makensie E. Millar, Eliese A. Potocek, Helen J. Wing

**Affiliations:** School of Life Sciences, University of Nevada Las Vegas, Las Vegas, NV 89154-4004, USA; monika.karney@unlv.edu (M.MA.K.); joy.immak@unlv.edu (J.A.M.); nweatherspoon@gmail.com (N.W.G.); Alexander.Karabachev@med.uvm.edu (A.D.K.); makensiem@gmail.com (M.E.M); potocek.e@gmail.com (E.A.P.)

**Keywords:** VirB, DNA-binding site, in vivo binding, *Shigella*, virulence, transcription, anti-silencing

## Abstract

The transcriptional anti-silencing and DNA-binding protein, VirB, is essential for the virulence of *Shigella* species and, yet, sequences required for VirB-DNA binding are poorly understood. While a 7-8 bp VirB-binding site has been proposed, it was derived from studies at a single VirB-dependent promoter, *icsB*. Our previous in vivo studies at a different VirB-dependent promoter, *icsP*, found that the proposed VirB-binding site was insufficient for regulation. Instead, the required site was found to be organized as a near-perfect inverted repeat separated by a single nucleotide spacer. Thus, the proposed 7-8 bp VirB-binding site needed to be re-evaluated. Here, we engineer and validate a molecular tool to capture protein-DNA binding interactions in vivo. Our data show that a sequence organized as a near-perfect inverted repeat is required for VirB-DNA binding interactions in vivo at both the *icsB* and *icsP* promoters. Furthermore, the previously proposed VirB-binding site and multiple sites found as a result of its description (i.e., sites located at the *virB*, *virF*, *spa15*, and *virA* promoters) are not sufficient for VirB to bind in vivo using this tool. The implications of these findings are discussed.

## 1. Introduction

In *Shigella* species, expression of a set of virulence genes carried on the large virulence plasmid, pINV (~220 kb), is controlled by transcriptional silencing and anti-silencing. The DNA-binding protein, VirB, is central to transcriptional anti-silencing and, therefore, essential for the virulence of these bacterial human pathogens; without VirB, *Shigella flexneri* is avirulent [1]. Upon VirB production at 37 °C [2], VirB binds DNA in a sequence-specific manner at VirB-regulated promoters to counter transcriptional silencing mediated by H-NS, the histone-like nucleoid structuring protein [3,4,5,6,7,8]. Despite the importance of VirB-DNA interactions, the DNA recognition sequence for VirB is poorly understood. 

Currently, our understanding of DNA sequences required for VirB binding stems from in vitro analyses of VirB binding at two VirB-dependent promoters: *icsB* [6,9] and *icsP* [7,10]. Since VirB belongs to the ParB protein superfamily [11], the putative VirB-binding site at the *icsB* promoter was initially identified due to its similarity to the Box A component of the binding site for P1 ParB (5′-(A/G)(A/T)G(G)AAAT-3′) [9]. Despite this sequence being organized as an inverted repeat, only the Box A-like sequence, subsequently renamed Box 2, was shown to be required for VirB-dependent regulation of the *icsB* promoter and important for VirB-DNA binding in vitro [6,9]. In contrast, at the *icsP* promoter, which is also organized as a near-perfect inverted repeat 5′-ATTTCAGtATGAAAT-3′, both boxes (Box 1 and Box 2) are required for VirB-dependent regulation [10]. 

In vitro investigations of the VirB-binding site [6,7,12] are complicated by the intrinsic property of VirB to form higher-order oligomers both on [7,12] and off DNA [13], leading to the production of non-discrete shifts in electrophoretic mobility shift assays [12] and long regions of protection in DNase I protection assays [7]. As such, if we are to re-analyze DNA sequences required for VirB binding, we reasoned that in vivo approaches would likely facilitate and lead to a greater understanding of these key regulatory sequences for *Shigella* virulence. Thus, we designed a genetic tool to elucidate DNA sequences required for VirB binding in vivo. 

The design of our genetic tool was inspired by the observation that the global transcriptional regulator of anaerobiosis, FNR, can repress a promoter by simple occlusion of RNA polymerase when an FNR-binding site is positioned near and within promoter elements [14]. In our genetic tool, termed the binding tool, protein-DNA binding is captured in vivo by placing a putative DNA binding sequence immediately upstream of the −35 promoter element of the constitutively active promoter, P*tac* [15], which controls the expression of a *lacZ* reporter. If the DNA-binding protein of interest binds the putative recognition site, RNA polymerase will be occluded from the promoter leading to reduced promoter activity and, consequently, lowered *lacZ* expression. 

In this study, we create and validate the utility of our binding tool using in vivo and in vitro approaches. We next use the tool and its derivatives to interrogate previously described VirB-binding sequences and investigate the components necessary and sufficient for VirB to bind to DNA in vivo. Our findings lead us to challenge the current understanding of a VirB-binding site and allow us to redefine the sequences necessary and sufficient for VirB-DNA binding. 

## 2. Materials and Methods

### 2.1. Bacterial Strains, Plasmids, and Media

The bacterial strains and plasmids used in this study are listed in Table 1. The isogenic *virB*::Tn5 *S. flexneri* 2a strain AWY3 [8] was routinely grown at either 25 °C or 37 °C in LB (Luria–Bertani) broth [16] with aeration,325 rpm in a LabLine/Barnstead 4000 MaxQ shaker (Thermo Fisher Scientific, Waltham, MA, USA) or plated on trypticase soy agar (TSA; containing 1.5% *w v*^−1^ agar). To ensure plasmid stability, all pBT series were stocked in *Escherichia coli* strain JM109(DE3) (Promega, Madison, WI, USA), which carries *lacI*^q^ needed to repress P*tac* under the standard growth conditions described above. For selection, antibiotics were added at the following final concentrations: ampicillin (100 μg mL^−1^), chloramphenicol (25 μg mL^−1^), kanamycin (50 μg mL^−1^), or carbenicillin (100 μg mL^−1^).

### 2.2. Construction of the In Vivo Binding Tool, pBT-Empty

The *tac* promoter, P*tac* [15], was polymerase chain reaction (PCR) amplified from pQE-60 (Qiagen, Germantown, MD, USA) with primers W344 and W345 (all primers listed in Table 2). The resulting amplicon was digested with SpeI and SalI, inserted into pBlueScript II KS(+) (Stratagene/Agilent, Santa Clara, CA, USA), and verified by Sanger sequencing (Genomics Core Facility at the University of Nevada Las Vegas, Las Vegas, Nevada, USA). Next, the *lacZ* reporter gene from pQF50 was introduced downstream of P*tac*. To do this, *lacZ* was PCR amplified from pQF50 [18] using W342 and W343. The resulting amplicon was digested with BamHI and XhoI, inserted into the plasmid backbone, and the Lac+ phenotype of cells bearing this construct was confirmed on plates containing X-gal (40 μg mL^‒1^). To limit large fluctuations in β-galactosidase activities, the P*tac-lacZ* was moved into the low-copy number plasmid, pACYC177 [17]. To do this, P*tac-lacZ* was removed from its plasmid backbone by digest with BsaAI and EcoO109I. The resulting fragment was ligated to pACYC177 that had been digested with DraIII, treated with T4 polymerase to create blunt ends, and subsequently digested with EcoO109I. A unique BglII site that would allow the insertion of putative binding sites immediately adjacent to the −35 element, was created by PCR-based site-directed mutagenesis using mutagenic oligonucleotides W395 and W396. The resulting amplicon was digested with SpeI and HindIII and used to replace the existing sequence. To remedy sequencing problems caused by a DNA repeat (generated during plasmid construction), an internal fragment of *gfp* was PCR amplified from pHJW43 using W412 and W413 and used to replace the repeated sequence. The resulting plasmid, pBT-empty, was confirmed by digest with BsrGI and NgoMIV and further verified by Sanger sequencing. pBT-empty has two *lacO* sites (5′-TTGTGAGCGGATAACAA-3′; [20] one overlapping P*tac* (−31 to −15) and the other downstream (+3 to +19)) that were used to repress P*tac* activity and prevent *lacZ* expression under standard growth conditions; nevertheless, neither of these sites impacted our in vivo binding assays because *S. flexneri* strains do not carry *lacI*.

### 2.3. Construction of the In Vivo Binding Tool Derivatives 

To construct the pBT derivatives, oligonucleotides were routinely annealed to create a 25 bp insert containing sites of interest flanked by native DNA sequences (unless otherwise noted) and compatible BglII and BamHI cohesive ends. Annealing reactions were done using an equimolar ratio of oligonucleotides, diluted to a final concentration of 1 pmol µL^−1^ in a Tris buffer (10 mM Tris, 1 mM EDTA, 50 mM NaCl (pH 7.5)). Oligonucleotide mixtures were annealed in a PCR machine [95 °C for 5 min, 1 °C less min^-1^ until annealing temperature reached, 30 min at annealing temperature, 1 °C less min^−1^ until 4 °C reached]. The 25 bp inserts were then ligated into the BglII site positioned immediately upstream of the −35 promoter element of P*tac* in pBT-empty. The forward orientation of the 25 bp insert was determined by digest with BglII and BamHI. All of the resulting constructs were confirmed by Sanger sequencing. To create the pBT derivative pNWG08 (DNase I protection assays), the P*tac* region carrying the VirB-binding site from the *icsP* promoter [10] was cut from pBT-P*icsP* with NotI and BamHI, and ligated to a similarly digested pBlueScript II KS(+). 

### 2.4. Purification of VirB

The VirB-His_6_ translational fusion protein was produced from pAJH03 and purified by the Monserate Biotechnology Group (San Diego, CA, USA) as described previously [7]. The His_6_-tag did not interfere with VirB expression or activity since His_6_-tagged VirB restored *icsP* expression to wild-type levels in a strain lacking *virB* in vivo [7]. 

### 2.5. DNase I Protection Assays

To identify regions bound by VirB and the RNA polymerase holoenzyme on pBT-*icsP* promoter fragments, DNase I protection assays were completed using [γ^−32^P]-ATP 5′ end labeled NotI BamHI fragments (159 bp). For each reaction, a final concentration of 6.4 nM of template DNA, sourced from pNWG08, was used with purified His_6_-tagged VirB (0, 30.7, 61.4, 122.8 or 307 nM) or *E. coli* RNA polymerase holoenzyme (0.0125, 0.025, 0.05, or 0.1 units µL^−1^; Product No. 78040Y supplied by Affymetrix, Inc/Thermo Fisher Scientific, Waltham, MA, USA). Samples were incubated at 37 °C for 30 min in a 20 μL reaction containing 50 mM Tris-HCl (pH 8.0), 20 mM KCl, 10% glycerol, 100 µg mL^−1^ bovine serum albumin (BSA), and 25 µg mL^−1^ poly (dI-dC). DNase I digestion was carried out as described previously [7]. Samples were treated with 3μl of 1:100 diluted DNase I (New England Biolabs, Ipswich, MA, USA) for 30 s followed by phenol-chloroform DNA extraction and ethanol precipitation. DNA was resuspended in a gel loading buffer (40% deionized formamide, 5 M urea, 5 mM NaOH, 1 mM EDTA, 0.025% bromophenol blue, and 0.025% xylene cyanol) and analyzed by 6% denaturing PAGE by comparing with a DNA sequence ladder generated using a Maxam and Gilbert A+G reaction [21]. Radiolabeled DNA was detected using a Typhoon 9410 (Amersham/GE, Chicago, IL, USA) variable mode imager and ImageQuant 5.2. 

### 2.6. β-Galactosidase Assays

To determine sequences required or sufficient for VirB binding, plasmids in the pBT series were introduced into the *S. flexneri virB* mutant strain AWY3 carrying the L-arabinose inducible pBAD-*virB* (pHJW16) or pBAD-empty control (pHJW14) by electroporation. The use of this inducible system allowed optimal assay conditions to be found given the low, but multi-copy nature of the pBT series. Promoter activities were determined by measuring β-galactosidase activity (protocol adapted from [22] and is described in [8]). Cultures were grown overnight (16 h) at 25 °C in LB broth containing 0.2% D-glucose, ampicillin (100 μg mL^−1^), and chloramphenicol (25 μg mL^−1^). Overnight cultures were then diluted 1:100 to grow for 8 h at 37 °C with aeration (325 rpm in a LabLine/Barnstead 4000 MaxQ shaker) in LB broth containing 0.2% L-arabinose, ampicillin (100 μg mL^−1^), and chloramphenicol (25 μg mL^−1^) prior to cell lysis. Three separate β-galactosidase assay trials, each from three independent transformations, were used to generate all presented data. Only the representative data is shown. The equality of variance for each dataset was calculated using F-tests and standard error *p* > 0.001 was calculated using an unpaired two-tailed Student’s *t*-test that assumed either equal or unequal variance. 

## 3. Results

### 3.1. Experimental Validation of the In Vivo Binding Tool

The starting point for this work was the newly constructed low-copy reporter plasmid, pBT-empty, which contains P*tac* [15] transcriptionally fused to *lacZ* (Figure 1Ai). To test DNA sequences required for VirB binding in vivo, putative cognate DNA sequences were inserted into a unique BglII restriction site positioned immediately upstream and adjacent to the −35 promoter element of P*tac* in pBT-empty. We hypothesized that if VirB binds to the inserted sequences, RNA polymerase would be sterically hindered from engaging promoter elements (Figure 1Aii) required for transcription initiation, which would result in low P*tac* activity and low *lacZ* expression [14]. However, if no sequences were inserted or the putative binding site was not sufficient for protein-DNA binding, RNA polymerase would not be occluded from promoter elements and *lacZ* would be expressed. 

To test the hypothesized function of our in vivo binding tool, we measured the P*tac* activities of pBT-empty and pBT-P*icsP*, which contains the VirB-binding site from the *icsP* promoter (Table 1), in the presence and absence of VirB (Figure 1B). To do this, pBT-empty or pBT-P*icsP* was introduced into *S. flexneri virB* mutant cells along with an L-arabinose inducible pBAD expression vector carrying either *virB* (pBAD-*virB*) or no gene (pBAD-empty). The P*tac* activities were measured using β-galactosidase assays. Our data show that under inducing conditions, P*tac* activity only decreases significantly when both VirB and the *icsP* VirB-binding site (pBT-P*icsP*) are present (Figure 1B). Additionally, in the absence of this VirB-binding site (pBT-empty) and regardless of whether VirB is present, P*tac* activity is not significantly altered. Thus, these data support the proposed function of our binding tool, namely, that when VirB binds to its cognate DNA sequence in pBT-*icsP* (as described previously; [7]) promoter occlusion occurs, resulting in lower β-galactosidase activities. Similar results were obtained when these pBT-derivatives were assayed in wild-type *S. flexneri* and an isogenic *virB* mutant derivative; however, due to the low but multi-copy nature of our binding tool reporter, exogenous expression of *virB* generated higher fold-changes (4-8 fold versus a 3 fold-change) in our assays.

To further validate our tool, we next chose to examine if the VirB and RNA polymerase footprints at the P*tac* region for pBT-P*icsP* overlap. To do this, regions of DNA bound by VirB and the RNA polymerase holoenzyme were identified using DNase I protection assays on a radiolabeled DNA fragment obtained from pBT-P*icsP*. DNA regions protected by VirB were observed with increasing VirB concentrations as compared to the no VirB control (Figure 1C). High affinity VirB-DNA binding (black line; Figure 1C) was observed between −46 to −69 relative to the *tac* transcription start site (+1), which correlates well with the inserted VirB-binding site positioned at −46 to −60 relative to the +1. Strikingly, with just a 2.5-fold increase in VirB (122.8 to 307 nM), an extended footprint was observed (dotted line; Figure 1C), once again revealing that with concentrations modestly higher than those needed to occupy the VirB binding site, VirB is able to spread along DNA sequences from its cognate site [7]. When investigating the DNase I protection exhibited by RNA polymerase holoenzyme, DNA sequences between +22 to −37 (black line; Figure 1C) were found to be strongly protected at all concentrations used. Notably, however, even with the lowest concentration used, partial protection of sequences located between −55 to −71 (dotted line; Figure 1C), a region that overlaps the VirB-protected region (−69 to −46), was observed. This demonstrates that VirB and RNA polymerase bind to overlapping DNA sequences in this semi-synthetic promoter. Altogether, these data support our conceptual model of the binding tool; that VirB binds to its cognate DNA sequence in pBT-*icsP* (as described previously; [7]) and occludes RNA polymerase from initiating transcription at P*tac*, and so we conclude that our binding tool can, indeed, be used to capture specific VirB-DNA binding interactions in vivo.

### 3.2. VirB Binds the Near-Perfect Inverted Repeats from the *icsP* and *icsB* Promoters In Vivo

Having validated the function of our in vivo binding tool, we next addressed the apparent discrepancy between the VirB-binding sites located at the *icsP* [7,10] and *icsB* promoters [6,9] (i.e., that VirB-dependent regulation requires only Box 2 at the *icsB* promoter, whereas the entire inverted repeat (Box 1 and Box 2) is required at the *icsP* promoter). To do this, we opted to test the role of both boxes from either the *icsP* or *icsB* promoters for VirB binding in vivo. Sequences of interest (i.e., native Box 1, Box 2, or the inverted repeats; Figure 2A,B) were inserted into pBT-empty and promoter activities of the resulting constructs were measured in a *S. flexneri virB* mutant harboring either pBAD-*virB* or pBAD-empty using β-galactosidase assays. 

Again, our results show that P*tac* activity from pBT-P*icsP*, containing the near perfect inverted repeat, was significantly decreased in the presence of VirB as compared to its absence, suggesting that VirB is able to bind and occlude RNA polymerase from P*tac* under these conditions (Figure 2C and Figure 1B). However, when Box 1, Box 2, or both are mutated (Figure 2A), no significant change in promoter activity was observed in the presence or absence of VirB (Figure 2C). In addition, when two additional nucleotides were inserted between the two boxes that comprise the inverted repeat (pBT-P*icsP*-2bp), a statistically significant but modest decrease in P*tac* activity was observed in the presence of VirB as compared to its absence (Figure 2C). Thus, we conclude that the presence of both boxes is not sufficient for VirB binding in vivo if the spacing is not optimal. 

For sequences taken from the *icsB* promoter (Figure 2B,D), our results showed that the P*tac* activity of pBT-P*icsB*, containing Box 1 and Box 2 from the *icsB* promoter, was significantly lower in the presence of VirB as compared to its absence (Figure 2D). These results were expected since the characterized VirB-binding site for the *icsB* promoter, Box 2 [6,9], was present. However, when Box 2 alone was present (i.e., Box 1 mutated; pBT-P*icsB*-mut1), surprisingly a decrease in P*tac* activity was not observed in the presence of VirB (Figure 2D). Similarly, when Box 1 alone was present (i.e., Box 2 mutated; pBT-P*icsB*-mut2) promoter activity was not altered in the presence or absence of VirB (Figure 2D). Thus, contrary to previous findings that report that Box 2 alone is necessary and sufficient for VirB binding at the *icsB* promoter, our results show that Box 1 and Box 2 are both required.

### 3.3. Re-Examination of VirB-Binding Sites Described in the Literature Using our In Vivo Binding Tool

The proposed VirB-binding site 5′-(A/G)(A/T)G(G)AAAT-3′ was developed during the investigation of VirB-dependent regulation of the *icsB* promoter [9] and subsequently used to search for putative binding sites within the VirB-dependent promoters of *virA, spa15, virF, and virB* [9,19]. Since our in vivo experiments (Figure 2) demonstrate that VirB only binds to this sequence when it is organized as an inverted repeat, we hypothesized that the currently proposed 7-8 bp VirB-binding site [9] and subsequent sites that were identified based on this sequence [9,19] would not be sufficient for VirB-DNA binding in vivo. To test this, annealed oligonucleotides carrying sequences of interest (Figure 3A) were inserted in the BglII restriction site in pBT-empty. The plasmid constructs were then introduced into a *S. flexneri virB* mutant, along with either pBAD-*virB* or pBAD-empty. VirB binding was captured in vivo by measuring P*tac* activity using β-galactosidase assays.

Our data show that in the presence of VirB, the currently proposed VirB-binding site (pBT-proposed) and putative sites established from it (i.e., sites from the *spa15* (pBT-P*spa15*), *virA* (pBT-P*virA*), *virB* (pBT-P*virB*), and *virF* (pBT-P*virF*) promoters [9,19]) did not significantly decrease P*tac* activity in the presence of VirB (Figure 3B), suggesting that none of these sites are sufficient for VirB binding in vivo, at least within the context of the binding tool. This is in stark contrast to the wild-type sequences taken from the *icsP* and *icsB* promoter, which were again found to be required for VirB binding in vivo (Figure 3B). Cumulatively, these results strongly suggest that VirB-DNA binding requires a near-perfect inverted repeat with the sequence 5′-ATTT(C)C(A/T)(C/T)n(A/G)(A/T)G(G)AAAT-3′. 

## 4. Discussion

In this work we describe and validate a new genetic tool that can be used to examine protein binding to specific DNA sequences in vivo. Simply by placing a putative binding site immediately adjacent to the −35 element of a constitutively active promoter, we were able to indirectly observe protein binding through promoter occlusion using β-galactosidase assays. We anticipate that this tool may prove helpful to those investigating DNA-protein interactions, especially when the DNA-binding protein is known to be recalcitrant in vitro.

Using our newly described tool, we examined DNA sequences previously reported to be required for the binding of a key transcriptional regulator of *Shigella* virulence genes, VirB. Strikingly, our data show that only sites organized as near perfect inverted repeats separated by a single nucleotide are required for VirB binding in vivo. This contrasts with reports in the literature that indicate that a single 7-8 bp site, constituting the currently proposed VirB-binding site 5′-(A/G)(A/T)G(G)AAAT-3′ [9], is sufficient for VirB binding. Our findings demonstrate that neither this proposed 7-8 bp site nor many of the subsequently identified putative sites (i.e., those described at the VirB-dependent *virB*, *virF*, *virA*, and *spa15* promoters; [9,19]) are sufficient for DNA binding in vivo. As such, this study challenges our current understanding of VirB-binding sequences and raises questions about these discrepant findings. 

Many of the experiments that led to the description of the currently proposed VirB-binding site 5′-(A/G)(A/T)G(G)AAAT-3′ were different from those used in this work. For instance, these foundational experiments were done at ambient temperatures rather than 37 °C, measured VirB-dependent regulation and not binding *per se*, and, perhaps more importantly, relied upon the exogenous expression of *virB* from an inducible plasmid well into stationary phase [6,9,19]. Although follow-up in vitro experiments were used to support the finding that VirB-dependent regulation requires a 7-8 bp VirB-binding site, we now know that these experiments have limited use when determining DNA sequence requirements for VirB binding because VirB is a promiscuous DNA-binding protein. Indeed, even in the aforementioned assays, VirB engaged DNA fragments bearing mutated VirB-binding sites, although higher concentrations of VirB were required to occupy mutated sites than wild-type sites [6]. 

In support of our finding that the bona fide VirB-binding site comprises an inverted repeat separated by a single nucleotide, in silico analysis of the large virulence plasmid of *S. flexneri* 2a, pCP301 [23] for the proposed 7-8 bp VirB-binding site, identified 258 perfect matches and 4,332 matches with a single nucleotide mismatch. Similar results were obtained when the large virulence plasmid of *S. flexneri* 5a, pWR100 [24] was analyzed. In contrast, only eight sites were found when we searched for our proposed VirB-binding site either as a perfect match or with a single mismatch. This is a more biologically relevant number given 10-15 promoters, which are estimated to control about 50 genes, are estimated to be regulated by VirB [24]. Intriguingly, however, these eight sites mapped to the intragenic regions of *ipaH7.8* and *spa47,* the intergenic regions of *ospD1* and *icsB* (and hence *ipgD*), both the intergenic and intragenic of *icsP*, upstream of a putative transposase, and in the *parS* locus of pINV. As such, these sites do not perfectly map to the promoters of VirB-regulated genes identified by high-quality macroarray analysis [24]. Work is in progress to address these discrepant findings. 

While we favor the idea that genuine VirB-binding sites are organized as an inverted repeat, it is possible that in some cases multiple 7-8 bp sites arranged in a modular format may support VirB binding. This kind of arrangement may explain why the 7-8 bp sites identified at the *virB*, *virF*, *virA*, and *spa15* promoters [9,19] were found to be insufficient for VirB binding when taken out of their natural context and inserted into our binding tool. If modular VirB-binding sites exist, however, then our work highlights that the correct spacing or phasing of these sites is likely to be important for VirB binding, because a 2 bp insertion between the two 7-8 bp sites that comprise the *icsP* inverted repeat significantly lowered VirB binding (Figure 2C). These ideas will be tested in future work. 

In conclusion, this study provides new insight into the DNA-binding requirements of a key regulator of virulence gene expression in *Shigella* species. Based on our findings, we suggest that VirB-binding sites are primarily organized as inverted repeats comprised of the half-sites 5′-(A/G)(A/T)GAAAT-3′ that are separated by a single nucleotide spacer. Examples of these inverted repeats are found at the *icsP* promoter (as reported previously; [7,10,25]) but also at the *icsB* promoter. While other examples of this site on the *Shigella flexneri* genome exist, their role in VirB binding and VirB-dependent regulation of proximal genes is currently being investigated. In closing, if we are to more thoroughly understand where VirB binds to DNA, the arrangement of these sites with respect to one another, and how these VirB-DNA interactions ultimately allow transcriptional anti-silencing of *Shigella* virulence genes, genome-wide studies that capture all VirB-DNA interactions will be required. 

## Figures and Tables

**Figure 1 genes-10-00149-f001:**
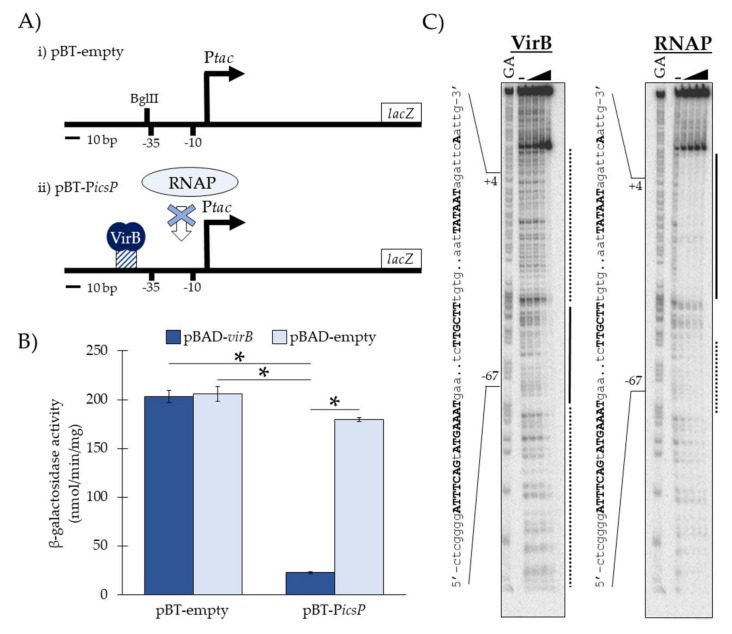
Schematic and validation of the in vivo binding tools pBT-empty and its derivative pBT-P*icsP*. (**A**) Schematic of (i) pBT-empty and (ii) pBT-P*icsP*, showing the hypothesized steric hindrance of RNA polymerase (RNAP) when VirB engages its binding site. (**B**) P*tac* promoter activity associated with pBT-empty or pBT-P*icsP* in the presence (pBAD-*virB*) or absence (pBAD-empty) of VirB in a *S. flexneri virB* mutant, as determined by β-galactosidase assays. Assays were conducted in triplicate, and representative data of three independent trials are shown. A Student’s *t*-test was used to measure statistical significance, * *p* < 0.001. (**C**) DNase I protection assays showing that VirB and RNAP occupy overlapping regions in pBT-P*icsP.* Lanes are organized (left to right): A+G sequencing ladder [21] and template DNA (6.4 nM) incubated with zero (-) or increasing final concentrations of His_6_-tagged VirB (30.7, 61.4, 122.8, or 307 nM) or RNA polymerase (0.0125, 0.025, 0.05, or 0.1 units µL^−1^). Black lines represent regions protected from DNase I at low concentrations (i.e., high affinity protein-DNA binding); whereas dotted lines represent regions protected from DNase I at high concentrations (i.e., low affinity protein-DNA binding). Bold and uppercase text are the VirB-binding site (inverted repeat), promoter elements (−35 and −10) and transcription start site (+1).

**Figure 2 genes-10-00149-f002:**
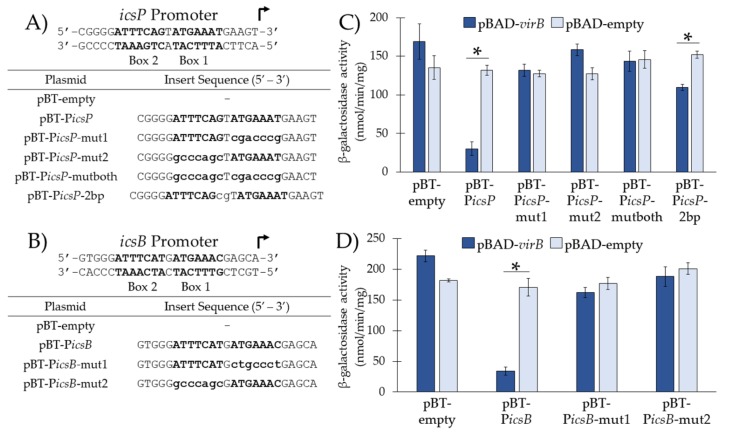
Investigation of sequences required and sufficient for VirB-binding in vivo from the *icsP* and *icsB* promoters. (**A**,**B**) DNA sequences inserted into pBT-empty. Uppercase nucleotides refer to native sequences; lowercase nucleotides refer to mutated sequences. Box 1 and Box 2 are in bold. (**C,D**) P*tac* activities were measured in the presence (pBAD-*virB*) or absence (pBAD-empty) of VirB in a *S. flexneri virB* mutant using β-galactosidase assays. Data is representative of three independent trials and * denotes a statistical significance of *p* < 0.001.

**Figure 3 genes-10-00149-f003:**
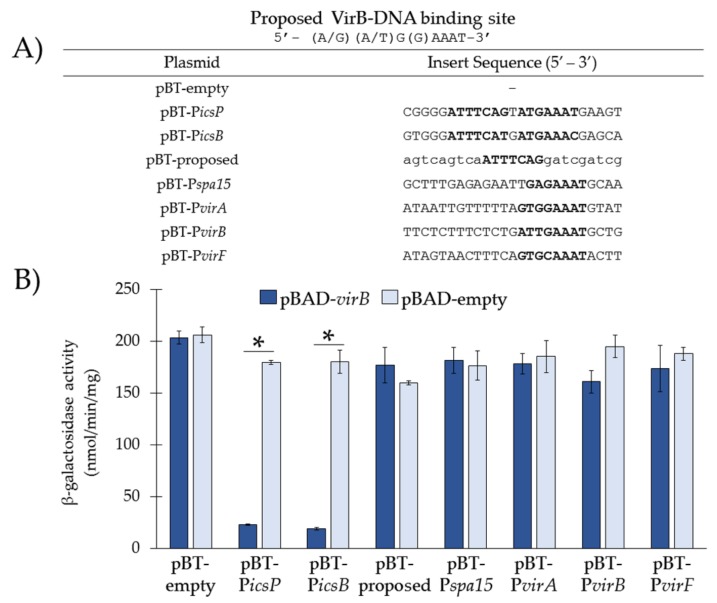
Re-examination of VirB-binding sites described in the literature using our in vivo binding tool. (**A**) DNA sequences inserted into pBT-empty. Putative VirB-binding sites described in the literature are in bold. Uppercase nucleotides refer to native sequences or the proposed site; lowercase nucleotides refer to scrambled sequences. (**B**) P*tac* activities were measured in the presence (pBAD-*virB*) or absence (pBAD-empty) of VirB in a *S. flexneri virB* mutant using β-galactosidase assays. Data is representative of three independent trials and * denotes a statistical significance of *p* < 0.001.

**Table 1 genes-10-00149-t001:** Bacterial strains and plasmids used in this study.

Bacterial Strain	Description	Source
AWY3	*Shigella flexneri* serotype 2a strain 2457T *virB*::Tn5; Kn^r^.	[8]
JM109(DE3)	*Escherichia coli* strain derived from JM109.	Promega
**Plasmids**	**Description**	**Source**
pACYC177	Low-copy number plasmid vector with a p15A ori; Amp^r^.	[17]
pQF50	Promoterless *lacZ* transcriptional fusion vector; Cb^r^.	[18]
pQE-60	Vector for C-terminally fused His_6_-tagged insert; Amp^r^.	Qiagen
pBlueScript II KS(+)	Multi-copy cloning vector; Amp^r^.	Stratagene
pHJW14	pBAD18; Cm^r^.	[8]
pHJW16	pHJW14-*virB*; Cm^r^.	[8]
pHJW43	Source of *gfp* used to create pBT-empty; Amp^r^.	This work
pAJH03	IPTG inducible plasmid carrying His_6_-tagged *virB.*	[7]
pNWG08	pBlueScript II KS(+) with *Not*I *BamH*I sequence of pBT-P*icsP*; Amp^r^.	This work
pBT-empty	pACYC177 carrying P*tac-lacZ* with a unique BglII site immediately adjacent to the −35 promoter element.	This work
pBT-P*icsP*	pBT-empty with P*icsP* Box 1 and 2 separated by a single nucleotide spacer 5′-ATTTCAGtATGAAAT-3′ [10]	This work
pBT-P*icsP*-mut1	pBT-empty with mutated P*icsP* Box 1	This work
pBT-P*icsP*-mut2	pBT-empty with mutated P*icsP* Box 2	This work
pBT-P*icsP*-mutboth	pBT-empty with mutated P*icsP* Box 1 and Box 2	This work
pBT-P*icsP*-2bp	pBT-empty with P*icsP* Box 1 and 2 with an added 2 bp spacer	This work
pBT-P*icsB*	pBT-empty with P*icsB* Box 1 and 2 separated by a single nucleotide spacer 5′-ATTTCATgATGAAAC-3′ [6]	This work
pBT-P*icsB*-mut1	pBT-empty with mutated P*icsB* Box 1	This work
pBT-P*icsB*-mut2	pBT-empty with mutated P*icsB* Box 2	This work
pBT-proposed	pBT-empty with proposed 7 bp VirB-binding site [9] flanked by scrambled sequences	This work
pBT-P*spa15*	pBT-empty with putative VirB-binding site 5′-GAGAAAT-3′ [9] from P*spa15*	This work
pBT-P*virA*	pBT-empty with putative VirB-binding site 5′-GTGGAAAT-3′ [9] from P*virA*	This work
pBT-P*virB*	pBT-empty with putative VirB-binding site 5′-ATTGAAAT-3′ [19] from P*virB*	This work
pBT-P*virF*	pBT-empty with putative VirB-binding site 5′-GTGCAAAT-3′ [19] from P*virF*	This work

**Table 2 genes-10-00149-t002:** Oligonucleotides used in this study.

Primer^1^	Sequence (5′ to 3′)	Description or Use
W342	ctagaggatccccgggtacccg	Amplification of *lacZ* from pQF50
W343	gcattactcgagttccttacgcgaaagacgggc	
W344	ggaaactagtcgattcgtaggccttgctttgtgagcggataac	Amplification of P*tac* from pQE60
W345	cgcggtcgaccccatatcaccagctcaccg	
W395	ctagaactagtcgattcgtagatcttgctttgtgagcgg	Mutagenic primers to generate unique B*gl*II site required for construction of pBT-empty
W396	agctagaagcttctagagatcccc	
W412	tgcattgcggccgcttttctgtcagtggagaggg	Amplification of ‘*gfp* required for construction of pBT-empty
W413	actgcagccggcaagaaggaccatgtgg	
W414	gatctCGGGGATTTCAGTATGAAATGAAGTg	Annealed to generate wild-type P*icsP* Box 1 & 2
W463	gatccACTTCATTTCATACTGAAATCCCCGa	
W527	gatctCGGGGATTTCAGTcgacccgGAAGTg	Annealed to generate mutated P*icsP* Box 1
W528	gatccACTTCcgggtcgACTGAAATCCCCGa	
W529	gatctCGGGGgcccagcTATGAAATGAAGTg	Annealed to generate mutated P*icsP* Box 2
W530	gatccACTTCATTTCATAgctgggcCCCCGa	
W416	gatctCGGGGgcccagcTcgacccgGAACTg	Annealed to generate mutated P*icsP* Box 1 & 2
W464	gatccAGTTCcgggtcgAgctgggcCCCCGa	
W587	gatctCGGGGATTTCAGcgTATGAAATGAAGTg	Annealed to create insert with 2 bp between P*icsP* Box 1 & 2
W588	gatccACTTCATTTCATacGCTGAAATCCCCGa	
W443	gatctTGCTCGTTTCATCATGAAATCCCACg	Annealed to generate wild-type P*icsB* Box 1 & 2
W444	gatccGTGGGATTTCATGATGAAACGAGCAa	
W445	gatctTGCTCagggcagCATGAAATCCCACg	Annealed to generate mutated P*icsB* Box 1
W446	gatccGTGGGATTTCATGctgccctGAGCAa	
W474	gatctTGCTCGTTTCATCgctgggcCCCACg	Annealed to generate mutated P*icsB* Box 2
W450	gatccGTGGGgcccagcGATGAAACGAGCAa	
W596	gatctagtcagtcaATTTCAGgatcgatgcc	Annealed to generate proposed VirB-binding site
W597	gatccCGATCGATCCTGAAATTGACTGACTa	
W627	gatctGCTTTGAGAGAATTGAGAAATGCAAg	Annealed to generate putative VirB-binding site at P*spa15*
W628	gatccTTGCATTTCTCAATTCTCTCAAAGCa	
W629	gatctATAATTGTTTTTAGTGGAAATGTATg	Annealed to generate putative VirB-binding site at P*virA*
W630	gatccATACATTTCCACTAAAAACAATTATa	
W631	gatctTTCTCTTTCTCTGATTGAAATGCTGg	Annealed to generate putative VirB-binding site at P*virB*
W632	gatccCAGCATTTCAATCAGAGAAAGAGAAa	
W633	gatctATAGTAACTTTCAGTGCAAATACTTg	Annealed to generate putative VirB-binding site at P*virF*
W634	gatccAAGTATTTGCACTGAAAGGGACTATa	
W119	gccagggttttgggagtcacga	Sequencing primer, M13F
W120	gagcggataacaatttcacacagg	Sequencing primer, M13R
W482	tgtggtgcaacgggcgctgg	Sequencing primer
W483	agaagcctgcgatgtcggtt	Sequencing primer
W484	caccgatattatttgcccga	Sequencing primer
W485	cctctggatgtcgctccaca	Sequencing primer
W486	ggcagcatcaggggaaaacc	Sequencing primer
W487	gcacatttccccgaaaagtg	Sequencing primer
W488	ggaagacgtacggggtatac	Sequencing primer
W489	ccagctcgatgcaaaaatcc	Sequencing primer
W490	ccacccagtcccagacgaag	Sequencing primer
W491	ccacagcggatggttcggat	Sequencing primer
W492	cggtttatgcagcaacgaga	Sequencing primer
W541	gcggccgctctagaactagtcg	Sequencing primer

The VirB-binding site components, Box 1 and 2, are indicated in bold. DNA sequences native to the *icsB*, *icsP*, *spa15*, *virA*, *virB*, and *virF* promoters are in uppercase and mutated/random sequences are in lower-case. All oligonucleotides were ordered from Integrated DNA Technologies^1^.

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
