# Peer review of "Investigating the DNA-Binding Site for VirB, a Key Transcriptional Regulator of Shigella Virulence Genes, Using an In Vivo Binding Tool"

_genes, 2019, doi:10.3390/genes10020149_

Round 1

Reviewer 1 Report

Shigella flexneri is a Gram negative gastrointestinal pathogen that is closely related to E. coli. Virulence for S. flexneri is dependent on a large plasmid that encodes a Type 3 Secretion System as well as additional virulence factors. Regulation of the genes encoded by the virulence plasmid has been the topic of intense research and is an elegant example of transcriptional hierarchy, with 3 activators responding to different physiological cues. First, the host temperature of 37oC leads to production of VirF. VirF in turn, activates the transcription of VirB. VirB activates expression of genes required for entry as well as additional substrates of the Type 3 Secretion Apparatus (T3SA). When the T3SA becomes active, an effector chaperone (IpgC) is free to act as a coactivator with an AraC-class activator MxiE.

The black box in this elegant model is VirB. Whereas VirF and MxiE fall into line with families of canonical transcription factors, VirB looks most like some sort of partitioning factor. The biochemical studies that have been performed on VirB have been hampered by the fact that the protein doesn’t behave well (oligomerizes) and that some of these studies were carried out under non-physiological conditions. In the current study Wing and coworkers use a relatively simple genetic approach to address the question of VirB binding in vivo. The authors create a system where promoter elements that confer VirB-dependent regulation are moved in front of a strong ptac promoter. When VirB is present the reporter lacZ gene is silenced, basically placing a synthetic operator in front of the ptac promoter. The authors then carry out a series of classic “promoter bashing” experiments to delineate a new consensus binding site for VirB. This “revised” VirB element is an inverted repeat and is quite different from the previously proposed VirB-responsive motif. The assay seems to work well, the use of lacZ is always nice as it gives qualitative and quantitative results. The results of their initial reporter assays is corroborated by DNA footprinting.

Overall the manuscript is well written although the telling of the story becomes confusing in the second half. Part of this reviewer’s confusion is due to the use of the term “putative VirB-binding site” (line 233) which refers to a motif proposed by Watanabe and coworkers. And, if I have it right, the pBT-“proposed” is also the previous site. The data is convincing but it would be very helpful to have a sentence or two at the end of the results, in more active voice, explaining what a VirB element is likely to be instead of what it isn’t.

I would also like to have an expanded comparison of the work of Le Gall and coworkers that performed high-quality microarray analysis of genes on the virulence plasmid. Le Gall’s study identified the VirB-regulated genes to be: icsP,phoN2 ,virA, ospD2ospForf13ospD1ospC234,ospC1orf81ipaJorf137 and icsA. Do these genes include the “revised” VirB element?

Minor comment.

Can the authors please come up with a better name for their assay? It seems to work and is straightforward, “binding tool” is pretty generic and not very descriptive.

Author Response

Reviewer 1:

Shigella flexneri is a Gram negative gastrointestinal pathogen that is closely related to E. coli. Virulence for S. flexneri is dependent on a large plasmid that encodes a Type 3 Secretion System as well as additional virulence factors. Regulation of the genes encoded by the virulence plasmid has been the topic of intense research and is an elegant example of transcriptional hierarchy, with 3 activators responding to different physiological cues. First, the host temperature of 37oC leads to production of VirF. VirF in turn, activates the transcription of VirB. VirB activates expression of genes required for entry as well as additional substrates of the Type 3 Secretion Apparatus (T3SA). When the T3SA becomes active, an effector chaperone (IpgC) is free to act as a coactivator with an AraC-class activator MxiE.

 The black box in this elegant model is VirB. Whereas VirF and MxiE fall into line with families of canonical transcription factors, VirB looks most like some sort of partitioning factor. The biochemical studies that have been performed on VirB have been hampered by the fact that the protein doesn’t behave well (oligomerizes) and that some of these studies were carried out under non-physiological conditions. In the current study Wing and coworkers use a relatively simple genetic approach to address the question of VirB binding in vivo. The authors create a system where promoter elements that confer VirB-dependent regulation are moved in front of a strong ptac promoter. When VirB is present the reporter lacZ gene is silenced, basically placing a synthetic operator in front of the ptac promoter. The authors then carry out a series of classic “promoter bashing” experiments to delineate a new consensus binding site for VirB. This “revised” VirB element is an inverted repeat and is quite different from the previously proposed VirB-responsive motif. The assay seems to work well, the use of lacZ is always nice as it gives qualitative and quantitative results. The results of their initial reporter assays is corroborated by DNA footprinting.

We thank the qualified reviewer for their time and positive review of our manuscript. We found this review to be very helpful. It has allowed us to improve our manuscript prior to its publication. Below, we have addressed each of the critiques raised and describe how we have changed the manuscript, accordingly (line numbers are noted to facilitate review). We believe our manuscript is now ready for publication and hope that you will agree.

Overall the manuscript is well written although the telling of the story becomes confusing in the second half. Part of this reviewer’s confusion is due to the use of the term “putative VirB-binding site” (line 233) which refers to a motif proposed by Watanabe and coworkers. And, if I have it right, the pBT-“proposed” is also the previous site. The data is convincing but it would be very helpful to have a sentence or two at the end of the results, in more active voice, explaining what a VirB element is likely to be instead of what it isn’t.

We agree, the term “putative VirB-binding site” is confusing because we were using it to refer to two things: the proposed site (Watanabe et al.,) but also sites found as a consequence of the proposed site’s description, which were poorly validated. Consequently, we have replaced the term “putative VirB-binding site” with “VirB-binding sites described in the literature”, since this phrasing encompasses both of the aforementioned sites. We have also made this change in the Figure 3 legend.

You are correct, we agree that it is much better to end this section with what the site is, rather than what it is not. We have now added the following text at the end of the results section (line 255):

“Cumulatively, these results strongly suggest that VirB-DNA binding requires a near-perfect inverted repeat with the sequence 5’-ATTT(C)C(A/T)(C/T)n(A/G)(A/T)G(G)AAAT-3’.

I would also like to have an expanded comparison of the work of Le Gall and coworkers that performed high-quality microarray analysis of genes on the virulence plasmid. Le Gall’s study identified the VirB-regulated genes to be: icsP, phoN2, virA, ospD2ospForf13ospD1ospC234, ospC1orf81ipaJorf137 and icsA. Do these genes include the “revised” VirB element?

Thank you. This comment was raised by the other reviewer too. As such, we have chosen to include the following text in this paragraph of the discussion:

“Intriguingly, however, these eight sites mapped to the intragenic regions of ipaH7.8 and spa47, the intergenic regions of ospD1 and icsB (and hence ipgD), both the intergenic and intragenic of icsP, upstream of a putative transposase, and in the parS locus of pINV.  As such, these sites do not perfectly map to the promoters of VirB-regulated genes identified by high-quality macroarray analysis [24].  Work is in progress to address these discrepant findings.”

Minor comment.

Can the authors please come up with a better name for their assay? It seems to work and is straightforward, “binding tool” is pretty generic and not very descriptive.

We agree that the term “binding tool” is somewhat generic, but we define this tool in the “Introduction” (line 54);

 “In our genetic tool, termed the binding tool, protein-DNA binding is captured in vivo by placing a putative DNA binding sequence immediately upstream of the -35 promoter element of the constitutively active promoter, Ptac [15], which controls the expression of a lacZ reporter.”

So, we think it should be clear to our readers that the “binding tool” allows us to measure protein binding to DNA through promoter occlusion. Hence, we have opted to keep this terminology because it is simple and straight-forward.

Other changes made to the manuscript:

We realized, upon review, that there were some errors with our reference section. These have been corrected.

We also found that the PicsB site in Figure 3 was written incorrectly. This has been changed in the new version.

Reviewer 2 Report

Gene regulation in Shigella is a complicated beast involving in one part the interplay between the silencer HNS and the anti-silencer VirB. Still, however, as the authors tell us, this process is poorly defined. In particular, this study investigates the binding site of VirB using a novel developed tool in order to measure in vivo binding. This investigation is important to understand the binding requirements of this protein in vivo.

The paper is well written, with a clear description of the topic, and the gap in the literature that the authors are attempting to fill. The authors first validate their tool for use with VirB and their characterized site for IcsP, which they have shown requires both Box 1 and Box 2. This section is nicely described as is the methods, and this is an interesting tool with VirB stopping LacZ expression via an inhibition of RNA Polymerase binding. It would be interesting to see if this would work with other DNA binding proteins as an alternative method of studying binding sites and such work would have made this paper more significant. However, subsequent investigation of the icsB binding site, shows evidence that both Box 1 and Box 2 are required for VirB binding here also; which is  different to previous studies.  Furthermore, other putative binding sites  of additional regulated factors also show no evidence of binding, which questions the postulated VirB binding site. I think that the study is well described and adds to the literature and our understanding of virulence gene regulation in this important pathogen.

I do have a few queries for the authors.

1.      One quesiont I have is why the authors did not simply use a comparison between the wt and the virB mutant with presence of the plasmid pBT but rather used inducible VirB? The authors themselves say in the discussion the use of inducible VirB may modulate binding (line 273)? Why do you think this was not the case in your study? How did the authors determine the level of arabinose added and whether this was resulting in a comparative level of expression compared to the wt? Did they try a comparison between the wildtype 2457T and the VirB mutant using pBT-IcsP for instance?

2.      In the discussion, you describe data to support that the inverted repeat is required (both Box 1 and Box 2), starting at line 280. However, you do not provide a list of genes of interest. I feel like this data would be helpful to readers (rather than making them simply go and do the work themselves). For instance, are the inverted repeats located where they could regulate known VirB regulated genes ? You say that 10-15 promoters are regulated by VirB, are these 8 inverted repeats in the vicinity of these promoters?

Minor Point.

1.      Line 30    - reference 1 and 2 – I’m not sure that these are the most suitable references, perhaps adding the following reference would be useful (Adler, B., Sasakawa, C., Tobe, T., Makino, S., Komatsu, K., & Yoshikawa, M. (1989). A dual transcriptional activation system for the 230 kb plasmid genes coding for virulence-associated antigens of Shigella flexneri. Mol Microbiol, 3(5), 627-635.)

Author Response

Reviewer 2

We thank the qualified reviewer for their time and positive review of our manuscript. Your review helped us to improve our manuscript prior to its publication. We address each of your critiques below and describe how we have changed the manuscript (line numbers are included). We think our manuscript is now ready for publication and hope you agree.

Gene regulation in Shigella is a complicated beast involving in one part the interplay between the silencer HNS and the anti-silencer VirB. Still, however, as the authors tell us, this process is poorly defined. In particular, this study investigates the binding site of VirB using a novel developed tool in order to measure in vivo binding. This investigation is important to understand the binding requirements of this protein in vivo.

The paper is well written, with a clear description of the topic, and the gap in the literature that the authors are attempting to fill. The authors first validate their tool for use with VirB and their characterized site for IcsP, which they have shown requires both Box 1 and Box 2. This section is nicely described as is the methods, and this is an interesting tool with VirB stopping LacZ expression via an inhibition of RNA Polymerase binding. It would be interesting to see if this would work with other DNA binding proteins as an alternative method of studying binding sites and such work would have made this paper more significant.

Based on the fact that our newly created binding tool was inspired by the finding that an FNR-binding site will block promoter activity when it replaces an extended -10 promoter sequence, we fully expect that our new tool will  work with other DNA binding proteins, especially those that prove to be tricky to work with in vitro.   

However, subsequent investigation of the icsB binding site, shows evidence that both Box 1 and Box 2 are required for VirB binding here also; which is different to previous studies.  Furthermore, other putative binding sites of additional regulated factors also show no evidence of binding, which questions the postulated VirB binding site. I think that the study is well described and adds to the literature and our understanding of virulence gene regulation in this important pathogen.

I do have a few queries for the authors.

1.         One quesiont I have is why the authors did not simply use a comparison between the wt and the virB mutant with presence of the plasmid pBT but rather used inducible VirB? The authors themselves say in the discussion the use of inducible VirB may modulate binding (line 273)? Why do you think this was not the case in your study? How did the authors determine the level of arabinose added and whether this was resulting in a comparative level of expression compared to the wt? Did they try a comparison between the wildtype 2457T and the VirB mutant using pBT-IcsP for instance?

Good point! Yes, we have tried to use WT Shigella and an isogenic virB mutant for these assays and we saw similar results, although the fold change observed was approx. 3-fold as opposed to 4-8 fold with the currently described system. We put this down to the fact that our reporter is a low but multi-copy plasmid (10-15 copies per cell). Consequently, we think the inducible system just elevates levels of VirB so that all sites on our reporter plasmids are occupied giving a clearer effect.

It should be remembered that the key finding of this work is that VirB is not binding to sites that were previously described as VirB-binding sites, so we are not too concerned about the elevated levels of VirB we are using, If we were seeing more binding in our assays than previously reported that would be very different and give us pause.

To address this concern, we have revised our manuscript in the following ways:

On line 148-149, we had previously justified our use of exogenous VirB as, “This strategy was found to be optimal given the low, but multi-copy nature of the pBT series.”  However, we agree that this may not be as descriptive as it should be. Thus, we have changed this section to now read

The use of this inducible system allowed optimal assays conditions to be found given the low, but multi-copy nature of the pBT series.”

We have also chosen to add the following text to lines 182-184,

Similar results were obtained when our these pBT-derivatives were assayed in wild-type S. flexneri and a virB mutant derivative (data not shown), but due to the low but multi-copy nature of our binding tool reporter, exogenous expression of virB generated larger differences in our assays.”

2.      In the discussion, you describe data to support that the inverted repeat is required (both Box 1 and Box 2), starting at line 280. However, you do not provide a list of genes of interest. I feel like this data would be helpful to readers (rather than making them simply go and do the work themselves). For instance, are the inverted repeats located where they could regulate known VirB regulated genes? You say that 10-15 promoters are regulated by VirB, are these 8 inverted repeats in the vicinity of these promoters?

Thank you. This comment was raised by the other reviewer too. As such, we have chosen to include the following text in this paragraph of the discussion (lines 294-299):

“Intriguingly, however, these eight sites mapped to the intragenic regions of ipaH7.8 and spa47, the intergenic regions of ospD1 and icsB (and hence ipgD), both the intergenic and intragenic of icsP, upstream of a putative transposase, and in the parS locus of pINV.  As such, these sites do not perfectly map to the promoters of VirB-regulated genes identified by high-quality macroarray analysis [24].  Work is in progress to address these discrepant findings.”

Minor Point.

1.      Line 30    - reference 1 and 2 – I’m not sure that these are the most suitable references, perhaps adding the following reference would be useful (Adler, B., Sasakawa, C., Tobe, T., Makino, S., Komatsu, K., & Yoshikawa, M. (1989). A dual transcriptional activation system for the 230 kb plasmid genes coding for virulence-associated antigens of Shigella flexneri. Mol Microbiol, 3(5), 627-635.)

We agree.  As such, we have changed the previous reference 1 and 2 (Maurelli, 1984 & 1988) on line 30 to the reviewer’s recommendation (Adler, 1989).  This change is reflected in the reference list on lines 327-329 as well.  In addition, we moved the Maurelli, 1984 citation (previously reference 1) to line 31 where it is much more suitable. 

Other changes made to the manuscript:

We realized, upon review, that there were some errors with our reference section. These have been corrected.

We also found that the PicsB site in Figure 3 was written incorrectly. This has been changed in the new version.